# DIMENSION DEBATE: IS 3D A STEP TOO FAR FOR OPTIMIZING MOLECULES?

## ABSTRACT

The discovery of new molecular materials with desirable properties is essential for technological advancements, from pharmaceuticals to renewable energy. However, the discovery process is arduous, requiring many trial-and-error cycles of complex and expensive experiments. Bayesian optimization (BO) is commonly used to find and screen candidate molecules efficiently. However, it is unclear how to choose the right molecular representations for a Bayesian surrogate model: While molecules are 3-dimensional in nature, 3D features in BO have largely been underexplored. Indeed, 1D and 2D molecular features—which incur loss of information—are typically used. In this work, we study this discrepancy: Why have 3D features been overlooked for BO in materials discovery? To this end, we evaluate 3D features against standard lower-dimensional features. We assess their optimization performance on real-world chemistry datasets, considering both various settings such as low- & high-data regimes and transfer learning, and different types of Bayesian surrogates. This amounts to the evaluation of 35 different setups per dataset, totaling over 2100 distinct runs. Our large-scale work provides insights and modeling guides to chemists and practitioners on the trade-offs between 1D, 2D, and 3D representations, in a bid to further accelerate materials discovery.

## 1 INTRODUCTION

The discovery of new materials is crucial for technological advancements, yet experiments and simulations in chemistry are often expensive in terms of time and computational resources (Tom et al., 2024; van Mourik et al., 2014). To overcome this, Bayesian optimization (BO) has emerged as a method for efficiently exploring the vast space of potential materials and guiding experimental efforts toward the most promising candidates for calculations in computational chemistry (Korovina et al., 2020; Li et al., 2024; Ranković et al., 2024). BO relies on training probabilistic surrogate models, such as Gaussian processes (GPs) or neural networks (NNs), using feature vectors obtained from various representations of molecules.

In molecular discovery, how to best represent molecules remains unclear. Commonly, BO for materials discovery rely on simplified 1D and 2D representations (Felton et al., 2021; Griffiths et al., 2024; Häse et al., 2021), such as SMILES strings or 2D molecular graphs, which fail to capture the true 3D spatial and geometric complexities of molecules (Wigh et al., 2022). Recently, 3D GNNs have shown promise in extracting features from the geometric structure of molecules, while respecting known physical symmetries, potentially offering superior predictive performance (Batatia et al., 2022; Crivelli-Decker et al., 2024; Gilmer et al., 2017; Liao & Smidt, 2023). While interest in 3D representations is growing (Li et al., 2022), much of the existing research has not fully explored their applicability within BO for materials discovery, leaving a gap in understanding *why* 3D features—which is faithful to the 3D nature of molecules—are underused.

In this work, we address this question. We perform a comprehensive, systematic benchmark of 1D, 2D, and 3D molecular representations for materials discovery tasks. We leverage LLM-based feature extractor (Kristiadi et al., 2024) to handle 1D SMILES representation, 2D message-passing GNNs (Gilmer et al., 2017), and 3D equivariant GNNs (Liao & Smidt, 2023), combining them with popular probabilistic surrogates like GPs and Bayesian NNs. We assess their BO performances across various settings (e.g., transfer learning) and various data regimes. Using this setup, shown in Fig. 1, we compare these features-models-settings combination across 2100 runs to evaluate the

Figure 1: We investigate the effect of the number of dimensions (1D, 2D, 3D) in molecular representations for materials discovery. While common wisdom suggests that "more is better", 3D features are underused in BO. This leads to our main question: *"If 3D features are faithful to the 3D nature of molecules, why do nobody use them in BO for materials discovery?"*

trade-off between computational cost and predictive accuracy. We find that LLM methods consistently outperform, 2D methods yield better results to 3D in most cases, and transfer learning performs similarly to models trained on the specific property on the BO loop.

In sum, our contributions are as follows:

- Through a large-scale benchmarking effort, we empirically answer the question of *why* 3D molecular features are underused in various real-world materials discovery tasks.

- We investigate tradeoffs in leveraging different representation dimensionalities in various realistic settings, e.g. in transfer learning.

- We provide insights into the scalability and compatibility of different molecular feature extractors with popular choices of probabilistic surrogates.

All in all, our work provides guiding principles for chemists and practitioners in making practical modelling decisions. Code to reproduce our work can be found in the following anonymous repository: https://anonymous.4open.science/r/3D_Bayes-0F8F.

## 2 PRELIMINARIES

### 2.1 BAYESIAN OPTIMIZATION

BO addresses the problem of finding the global maximizer of an unknown objective function $f : \mathcal{X} \to \mathcal{Y}$. The goal is to find

$$\mathbf{x}_* = \arg\max_{\mathbf{x} \in \mathcal{X}} f(\mathbf{x}),$$

where $\mathcal{X}$ represents an arbitrary $d$-dimensional search space. Here, $f$ is assumed to be hard and expensive to evaluate. The challenge is, therefore, to minimize the number of such function evaluations when solving the above optimization problem (Shahriari et al., 2016). The key components of BO are (i) a *probabilistic belief* $p(f \mid \Omega_t)$ over the target function $f$ given past observation $\Omega_t = \{(\mathbf{x}_i, f(\mathbf{x}_i))\}_{i=1}^t$ and (ii) an *acquisition function* $\alpha : \mathcal{X} \to \mathbb{R}$, which guides where to evaluate $f$ next at step $t + 1$. The representational capacity of $p(f \mid \Omega_t)$ dictates how well-calibrated we can approximate $f$. This, in turn, affects the exploration-exploitation capabilities of the overall system, a critical factor in finding the optimal $\mathbf{x}_*$ in as few steps as possible (Garnett et al., 2012).

### 2.2 PROBABILISTIC SURROGATES

GPs are a common choice for the surrogate $p(f \mid \Omega_t)$ in BO (Shahriari et al., 2016), but alternatives like Bayesian NNs (BNNs) are also widely explored (Lamb & Paige, 2020).

**Gaussian processes** A GP can be seen as a distribution over functions $f(\mathbf{x})$, where for every finite collection $\{\mathbf{x}_i\}_{i=1}^t$, the distribution of $\{f(\mathbf{x}_i)\}_{i=1}^t$ is a multivariate Gaussian. In particular, this means $f(\mathbf{x}) \sim \mathcal{N}(f(\mathbf{x}) \mid \mu_\theta(\mathbf{x}), K_\theta(\mathbf{x}, \mathbf{x}))$ for some mean function $\mu_\theta$ and covariance function $K_\theta$ (Rasmussen & Williams, 2005). The covariance/kernel function effectively injects prior information about the function $f$ into our belief. Finally, $\theta$ contains the hyperparameters of the GP, which is often tuned via marginal-likelihood maximization.

**Bayesian neural networks** A neural network $f : \mathcal{X} \times \Theta \to \mathcal{Y}$, parameterized by $\theta \in \Theta$, is usually trained by maximizing the following *maximum a posteriori* (MAP) objective

$$\theta_* = \arg\max_{\theta \in \Theta} \log p(\Omega_t \mid \theta) + \log p(\theta) = \arg\max_{\theta \in \Theta} \log p(\theta \mid \Omega_t).$$

Since it is a point-estimation procedure, notice that this training scheme does not capture the uncertainty in $\theta$ and thus in $f$. Standard NNs are, therefore, unsuitable for BO. Various ways to construct a BNN exist, such as via variational inference (Blundell et al., 2015) and Markov chain Monte Carlo (Hoffman et al., 2014). One of the simplest, however, is the Laplace approximation (MacKay, 1992). It quantifies the uncertainty in $\theta$ by fitting a Gaussian distribution $p(\theta \mid \Omega_t) \approx \mathcal{N}(\theta_*, \Sigma_*^{-1})$, where the covariance matrix $\Sigma_*^{-1}$ is determined by the inverse Hessian of the negative log posterior $\Sigma_*^{-1} = -\nabla_\theta^2 \log p(\theta \mid \Omega_t)|_{\theta=\theta_*}$. To obtain the posterior over $f$, the linearized Laplace approximation (LLA) is often used (Daxberger et al., 2021), resulting in the Gaussian distribution over function outputs $\mathcal{N}(f(\mathbf{x}; \theta_*), J(\mathbf{x}; \theta_*)\Sigma_*^{-1}J(\mathbf{x}; \theta_*)^\top)$ where $J(\mathbf{x}; \theta_*) = \partial f(\mathbf{x};\theta)/\partial\theta|_{\theta=\theta_*}$. This distribution can then be used to model the target black-box function in BO (Kristiadi et al., 2023).

### 2.3 MOLECULAR FEATURE EXTRACTORS

Molecules can be expressed via various 1D, 2D, and 3D representations. These range from simple strings like SMILES to more complex graph-based and geometric descriptors. SMILES offers a 1D representation of molecular structures (Weininger, 1988). Graph-based representations leverage 2D graphs of molecular bonds (Duvenaud et al., 2015), and 3D representations capture the full geometric structure of molecules (Schütt et al., 2017). Before they can be used by a GP or BNN surrogate, these representations are often transformed into feature vectors. The following are various features that are commonly used in materials discovery.

**Molecular fingerprints** Molecular fingerprints are unique digital representations of molecular structures. They are generated by analyzing the atomic composition and connectivity within a molecule, resulting in a high-dimensional bit vector that uniquely identifies the molecule. One of the most commonly used are extended-circular fingerprints (Rogers & Hahn, 2010) which iterate over node centers and assign bits according to sub-structures up to 2 edges away (Weininger, 1988). Fingerprints capture structural information, such as the presence of specific functional groups or molecular motifs. They are widely used in cheminformatics for tasks like similarity searching, property prediction, and structure-activity relationship studies. In addition to fingerprints, other molecular descriptors are often employed to characterize physicochemical properties.

**Graph-neural-network features** Molecular fingerprints are computationally efficient but lack the structural richness required for complex tasks (He et al., 2021). Graph NNs (GNNs) have emerged as effective methods to capture local structures in graph-structured data, such as graph representations of molecules. Traditionally, GNNs are used to process 2D graphs by passing messages between their hidden layers to learn graph features (Duvenaud et al., 2015; Gilmer et al., 2017). Recently, equivariant GNNs that leverage the symmetries in 3D molecular structures, further improving the representational power of traditional GNNs, have been proposed (Liao et al., 2024).

**Large-language-model features** A significant advancement in the development of large neural networks is the introduction of the architecture called *transformer* (Vaswani et al., 2017), which serves as the foundation for large language models (LLMs). Due to the large size of LLMs, training them from scratch is computationally prohibitive (Sharir et al., 2020), but they are typically pre-trained in a task-agnostic manner, allowing them to serve as meaningful priors for various tasks (Brown et al., 2020). Recently, LLMs specifically tailored for chemistry-related applications have gained significant traction, particularly in their ability to extract meaningful features from chemical data (Chithrananda et al., 2020; Schwaller et al., 2019). These models are trained on large-scale chemical databases via

the molecules' 1D string representations. This makes LLMs useful as molecular feature extractors and has been used for materials discovery (Jablonka et al., 2023; Kristiadi et al., 2024).

## 2.4 3D FEATURES

3D molecular representations capture the full geometric structure of a molecule, providing critical information about atom positions, bond angles, and spatial orientation (Fang et al., 2022). This spatial configuration is particularly relevant for tasks where molecular conformation plays a role in determining chemical properties, such as catalysis or drug binding affinity (Turner et al., 2006). Historically, 3D features have been more computationally expensive than 1D or 2D representations, often requiring quantum mechanical calculations or molecular simulations to obtain accurate geometries (van Mourik et al., 2014). Methods such as SchNet (Schütt et al., 2017) and more recently, equivariant GNNs like EquiFormer (Liao & Smidt, 2023), have been developed to leverage these 3D structures by respecting their inherent symmetries, including rotational and translational invariance.

## 3 RELATED WORK

Various benchmarks have been developed for studying the effectiveness of BO surrogates in chemical applications. Olympus (Häse et al., 2021) offers a suite of common optimization response surfaces along with experimentally derived chemical prediction datasets for benchmarking. Similarly, Summit (Felton et al., 2021) offers a suite of virtual chemical reactions for optimization. Works have also studied and benchmarked the performance and uncertainty calibration of surrogate models on molecular screening tasks (Graff et al., 2021; Griffiths et al., 2024; Liang et al., 2021; Tom et al., 2023). Notably, these benchmarking works only consider (1D) string, molecular descriptors and fingerprints, and 2D graph representations of molecules. Our work complements these prior benchmarking efforts by investigating 3D representations that have largely been ignored.

Many sophisticated feature extractors have been used in BO for materials discovery. Strieth-Kalthoff et al. (2024) leverage graph features to accelerate the discovery of organic laser emitters by optimizing material properties across distributed labs in a closed-loop discovery system. However, they represent molecules as 2D graphs and use 2D GNNs as feature extractors. Kristiadi et al. (2024) and Liu et al. (2024) study transformers and LLMs as feature extractors in BO for materials discovery. While they represent the state-of-the-art general-purpose class of architectures, notably, they take 1D string representations of molecules (SMILES (Weininger, 1988), SELFIES (Krenn et al., 2020)) as inputs. Furthermore, there has been research on building graph foundation models (Zhou et al., 2023) utilizing molecular graph structures and self-supervised pretraining to achieve state-of-the-art performance in various chemistry-related tasks. Our work, meanwhile, studies the scenario where 3D molecular representations are used as the inputs of the feature extractor in material discovery tasks.

## 4 SETUP

We investigate BO performance using 1D, 2D, and 3D representations of molecules. This section outlines our experimental setup, including datasets, feature extractors, tasks, sample complexities, and evaluation methods.

**Datasets** We use four datasets in our experiments: QM7 (Blum & Reymond, 2009; Montavon et al., 2013), QM9 (Ramakrishnan et al., 2014; Ruddigkeit et al., 2012), GEOM's MoleculeNet and DRUGS (Axelrod & Gómez-Bombarelli, 2022; Wu et al., 2017). QM7 includes 7165 molecules with atomization energies ($\Delta$E) in kcal/mol and up to seven heavy atoms (C, N, O, S), while QM9 contains 133 885 molecules with up to nine heavy atoms (C, N, O, F) and 12 properties. The MoleculeNet dataset consists of benchmark datasets designed for molecular machine learning tasks and includes 28 295 molecules, covering tasks like quantum mechanics, physical chemistry, biophysics, and physiology. GEOM provides an enhanced version of this dataset that includes conformers for each example. On the other hand, GEOM also contains the DRUGS dataset, which provides molecular geometries for drug-like molecules, with up to 91 heavy atoms and 317 928 molecules, which are useful for studies involving conformational flexibility and geometric properties. The models were trained on QM9, which was split into a training set and a virtual library serving as the search space,

ensuring that the best 10 observations remain in the virtual library. The virtual library and all other datasets were used in the BO loop to evaluate the models.

**Tasks**   Each model is trained for target property prediction and transfer learning. In the target property prediction task, the model has a single readout layer trained to predict a specific property—e.g. HOMO-LUMO gap ($\Delta E_{\text{gap}}$ in eV) from QM9, crucial in determining molecular reactivity and optical properties; atomization energy ($\Delta E$ in kcal/mol) from QM7, used to understand molecular stability; and absolute energy ($E$ in Hartree) from GEOM's MoleculeNet and DRUGS, useful in studying binding energy and potential energy in the molecular system. The model has $n - 1$ readout layers in transfer learning, each trained on different tasks. We aim to assess whether a model trained on one set of properties can still provide accurate predictions on different experimental datasets by fine-tuning only the final layer, evaluating its potential as a foundation model (Yao et al., 2024).

**Feature extractors**   We compare MPNN, which inherently leverages 2D molecular information, Equiformer v2 (Liao et al., 2024), an equivariant attention-based GNN which captures the full 3D geometric structure and symmetries of molecules, and MolFormer (Kristiadi et al., 2024; Ross et al., 2022), a masked language model that operates on 1D SMILES representations, leveraging transformers to capture both local and global chemical patterns. The GNNs serve as feature extractors up to their respective readout layers, encoding molecules into high-dimensional embeddings before making predictions on the target properties. To ensure consistency, the GNN feature extractors are constrained to similar sizes, with each containing approximately 1.5 million parameters.

**Surrogates**   We use each features in two surrogate models. Each surrogate, either GP or an NN with LLA, uses the extracted features as inputs to provide a posterior distribution $p(f \mid \mathcal{D})$. The LLA surrogate consists of an NN with two hidden layers, as suggested by Li et al. (2024). As further baselines, we use (1) random search, which uniformly samples from the molecular space, and (2) GPs utilizing the Tanimoto kernel with 1D molecular fingerprints (Tripp et al., 2023).

**Evaluation**   We run 1000 iterations with 10 initial observations. For each run, we either subsample 10 000 observations from our virtual libraries (QM9, and GEOM's MoleculeNet and DRUGS) or use the entirety of the virtual library (QM7), repeating the processes for 15 different seeds, and reporting their average and standard error. We further evaluate with the GAP metric (Jiang et al., 2020), defined by GAP $= (y_i - y_0)/(y_* - y_0)$ where $y_i$ is the maximum observed value at step $i$ and $y^*$ is the true optimal value. Note that the GAP metric is normalized in $[0, 1]$ and is thus useful for comparing and aggregating results across different datasets/problems.

## 5   RESULTS

### 5.1   BAYESIAN OPTIMIZATION PERFORMANCE

**QM7**   For QM7, which features relatively simple molecules, highlights the surprising effectiveness of molecular fingerprints. Notably, the 1D GP method performed slightly worse than more complex models. In contrast, GP regression with binary encoded SMILES, serving as a baseline, demonstrated that even simple 1D representations can capture sufficient information to remain competitive. Although 2D models outperformed 3D models overall—particularly when combined with GP regressors—the performance gap was modest. While the 3D models saw slight improvements when paired with LLA, the gains were limited, suggesting that higher-dimensional representations may not be critical for simpler molecular structures like those in QM7. This is evident in Fig. 3, where the top-performing models for simpler tasks did not heavily rely on 3D data. The results of the LLM were striking, as it outperformed all other models by a significant margin. Its ability to leverage contextual information and generate accurate predictions, even for relatively simple molecules in the QM7 dataset, demonstrated its superior generalization capabilities.

**QM9**   In contrast, the QM9 dataset, which features slightly more complex molecules, underscores the limitations of 1D representations. Here, 2D MPNNs achieve the highest performance, and consistently outperform 1D GP and RS methods, while 3D models only outperform RS. The differences become even more pronounced with increasing molecular complexity as described by size. While 2D models continue to demonstrate strong performance and stability, the 3D GNNs, particularly

when enhanced with LLA, begin to close the gap, indicating that the extra structural information provided by 3D representations becomes more important as molecular complexity increases. Despite the overall advantage of the 2D models, the smaller margins between 2D and 3D performance suggest that for highly complex molecules, further optimization of 3D models may yield competitive results, as seen in the bottom row of Fig. 3. Contrary to all other datasets, LLMs performed worse than 2D and 3D models. This task may have been the most dependent on information not captured by 2D and 3D representations the specific, which could explain why it performed worse.

**MoleculeNet**   As shown in Fig. 3, 2D models consistently outperformed 3D models across a wide range of tasks, which suggests that 2D representations efficiently capture the necessary structural information for accurate predictions without the computational overhead of 3D models. The slight improvements observed with 3D models when using techniques like LLA are insufficient to justify their use, as the performance gains are marginal and insufficient. The consistently strong performance of 2D models raises important questions about the value of incorporating 3D information. Even as molecular size increases, 3D models fail to offer significant advantages, and in many cases, they underperform compared to 2D approaches. Additionally, the competitive performance of 1D models, such as GPs with SMILES encoding, highlights the efficiency of simpler representations in certain scenarios. Although 1D models struggle with larger datasets and more complex molecular structures, their ability to remain competitive in simpler tasks emphasizes that higher-dimensional representations are not always necessary. The LLM, as with the QM7 and DRUGS results, outperformed all other models. Its superior performance across both simple and complex molecular datasets highlights its ability to generalize effectively, surpassing the limitations of both 2D and 3D models. This reinforces the trend observed before, where the LLM demonstrated remarkable versatility and accuracy.

**GEOM DRUGS**   The DRUGS dataset emphasizes the importance of higher-dimensional features. However, despite the increased molecular size, both 1D and 2D representations manage to capture sufficient information to perform competitively. As shown in Fig. 3, LLMs and 2D models consistently outperform 3D models across most tasks, even for large drug-like molecules. Interestingly, simple GP regression and random search performed similarly, suggesting that the models used may not be sufficiently complex to outperform these benchmark methods. This indicates that larger models or further optimization would be necessary to see significant improvements beyond the baseline methods. The 3D models, while more suited for capturing subtle geometric features, do not provide a substantial performance increase, reinforcing that for property optimization in such datasets, 3D features may not be necessary unless ultra-high precision is required. The LLM achieved the most substantial performance gap so far, outperforming all other models by a wide margin. This is particularly remarkable given the complexity and size of the molecules in this dataset. Despite the strong performance of 2D models, the LLM's ability to handle intricate molecular details allowed it to excel far beyond both 2D and 3D representations. This suggests that the LLM's contextual understanding is especially beneficial for larger, more complex molecular structures.

**Aggregated results**   Contrary to the assumption that 3D features, which align more closely with the true nature of molecular structures, would offer superior performance, the aggregated results across all models from Fig. 2 consistently show that 1D and 2D representations outperform 3D approaches. Furthermore, regardless of molecular size, 1D representations such as SMILES proved surprisingly competitive. These models strike a balance between capturing essential molecular information and maintaining computational efficiency, making them highly effective for property prediction tasks in BO. On the other hand, 3D models, while expected to provide more detailed geometric insights, did not offer significant advantages over 2D models. The computational overhead

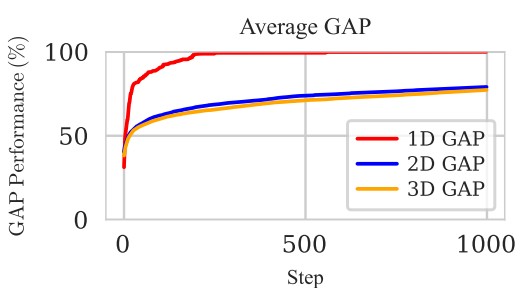

Figure 2: Aggregated GAP Performance for 1D, 2D and 3D models.

required by 3D models often outweighed their predictive performance, especially in cases where

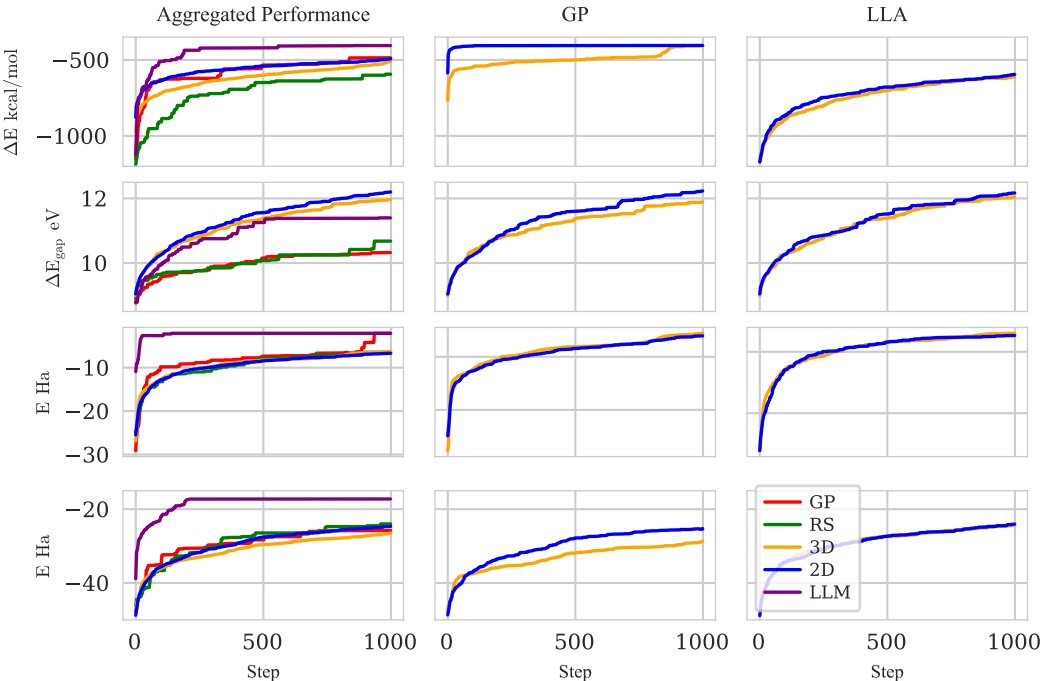

Figure 3: **Experimental Results**. Top row: QM7. Second row: QM9. Third row: GEOM's Molecule Net. Bottom row: GEOM's DRUGS.

molecular size was not extreme. Overall, the results suggest that 1D representations are a strong default choice for most BO tasks, providing strong performance without the additional complexity and cost of 3D features. This finding challenges the notion that higher-dimensional data always leads to better outcomes in molecular optimization, highlighting the practical utility of 1D or 2D models in real-world chemistry and materials discovery tasks.

> As molecular size increases, 1D models and 2D representations capture enough information to perform effectively, rendering 3D features generally capture enough information.

### 5.2 How many samples does each dimension need?

The models were trained on datasets with varying sample complexities to evaluate their performance based on the number of observations needed to utilize 3D information effectively. Previous research indicates that equivariant models generally require more samples than non-equivariant models to achieve similar performance levels (Elesedy & Zaidi, 2021). For the feature extractors, we experimented with four different training set sizes: 500, 1000, 10 000, and 50 000 observations. Thus, we investigated how model performance scales with sample size.

As illustrated in Fig. 4, 3D models consistently required a larger number of training samples to outperform or even match the performance of 2D models, particularly for simpler datasets such as QM7. In these lower-complexity tasks, the computational overhead introduced by 3D features did not translate into closes the performance gap until the sample size exceeded 10,000 observations. For example, while 3D models did show some improvement with more samples, their performance remained inferior to that of 2D models with smaller datasets.

In contrast, the 2D models were highly data-efficient across all datasets, capturing essential structural information with relatively few samples. Even with a modest dataset of 500 to 1,000 observations, 2D models achieved competitive performance, suggesting that the information content provided by 2D representations is generally sufficient for many molecular property prediction tasks. The gap

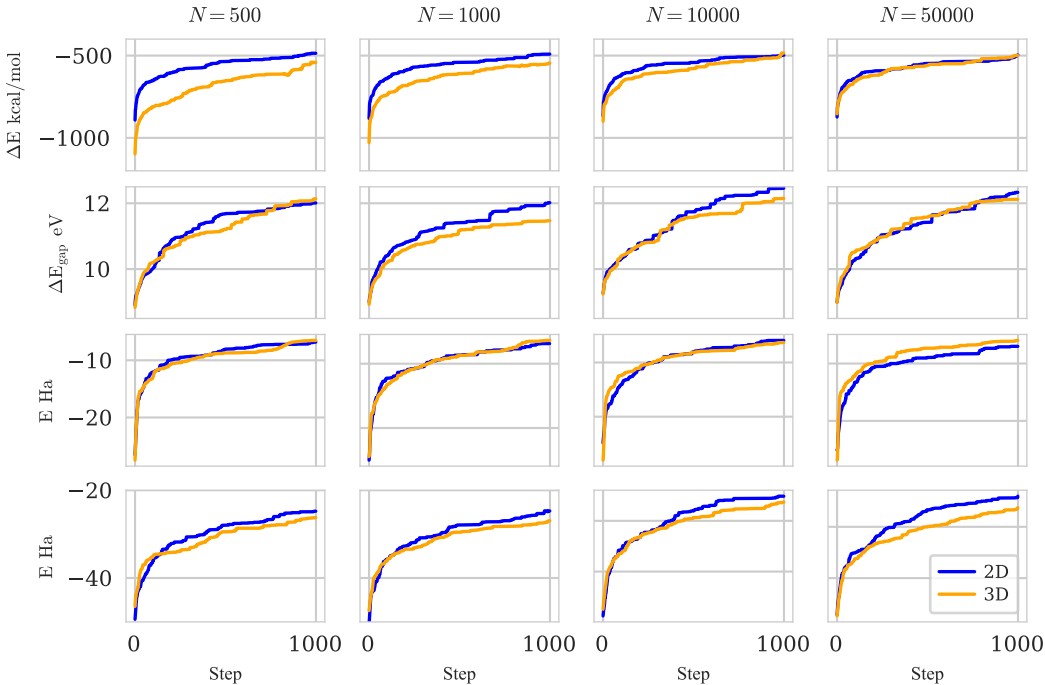

Figure 4: **Experimental Results per Sample Complexity**. Top row: QM7. Second row: QM9. Third row: GEOM's Molecule Net. Bottom row: GEOM's DRUGS.

between the performance of 2D and 3D models was most pronounced in smaller datasets, where 3D models often struggled to justify their computational expense. This finding aligns with earlier research (Elesedy & Zaidi, 2021), which highlights the difficulty of leveraging 3D information in data-scarce regimes.

> 3D models require more observations, particularly for simpler molecules, to match the performance of 2D models. In more complex datasets, the improvement of 3D models is minimal, indicating that 2D models are sufficient and more efficient.

### 5.3 IS TRANSFER LEARNING BENEFICIAL?

The results comparing 2D and 3D models across single-property prediction and transfer learning tasks, as shown in Fig. 5, reveal key differences in their effectiveness. The LLM used was trained in multiple tasks, thus offering only a transfer learning perspective. In single-property tasks, 2D models consistently outperform 3D models, particularly in datasets with limited data, like QM7 and QM9. This suggests that 2D representations capture essential structural information efficiently, without the computational cost of 3D models. Even in more complex datasets like GEOM DRUGS, where the performance gap between 2D and 3D models narrows, 2D models remain more competitive and effective for property prediction, offering a balance of simplicity and accuracy. However, in transfer learning—where models trained on one molecular property are fine-tuned to predict another—3D models show some improvement but still lag behind 2D models. The additional geometric detail provided by 3D representations enhances generalization across tasks but is not enough to outperform 2D models in terms of efficiency and accuracy.

Moreover, transfer learning appears to offer a viable path to generalizing across multiple molecular properties with minimal loss in accuracy, making it an attractive option for both 2D and 3D models. While 2D models continue to lead in terms of efficiency and precision, particularly in targeted tasks, the fact that transfer learning models can achieve performance levels close to task-specific models suggests that they can be a valuable tool for both types of models. Even though 3D models

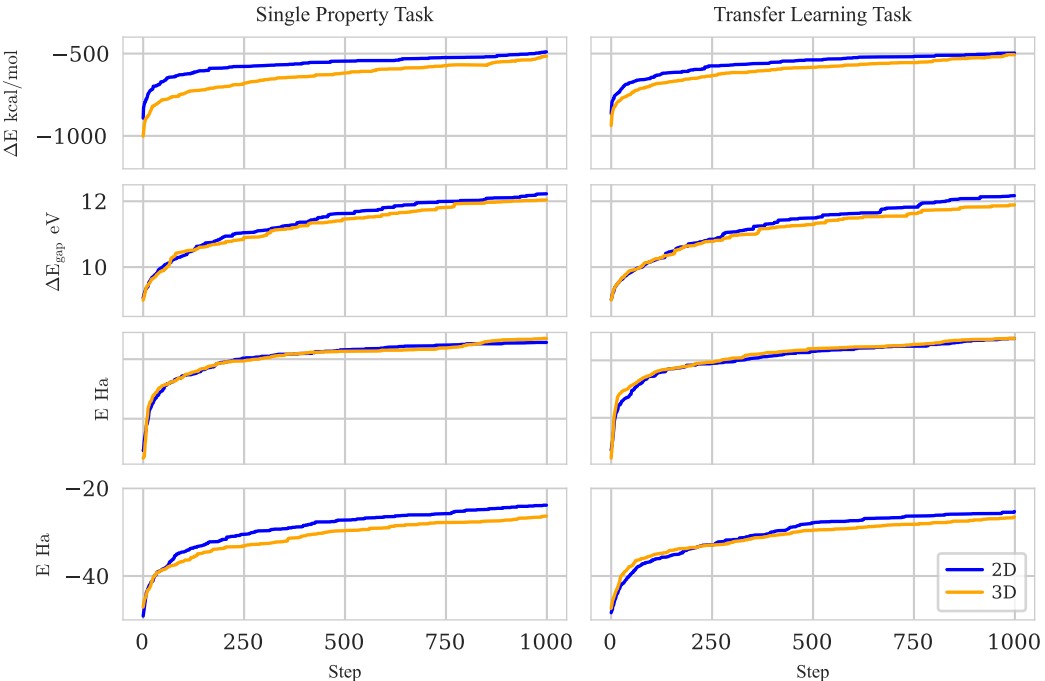

Figure 5: **Experimental Results per Task**. Top row: QM7. Bottom row: QM9.

do not outperform 2D models, they still show notable improvements through transfer learning, and this adaptability highlights the practicality of using transfer learning in scenarios that demand generalization across different molecular properties.

> Performance on transfer learning is close to that of specific task prediction. Foundation models prove a good tool to leverage in molecular optimization.

## 6 CONCLUSION

Across all datasets examined LLMs consistently outperformed both 2D and 3D models, offering a stable and highly efficient approach for molecular property prediction. This superiority was particularly evident in more complex datasets, such as GEOM DRUGS, where LLMs demonstrated a significant performance gap compared to other models, even for large, intricate molecules. Notably, 2D representations also showed promise, outperforming 3D models across a wide range of tasks. The relative success of 1D and 2D models highlights the advantage of simpler representations, which not only provide strong predictive accuracy but also strike an excellent balance between efficiency and computational cost. This advantage is particularly noticeable in datasets traditionally associated with 3D models, where 3D geometric information was expected to offer an edge but instead provided only marginal improvements at a higher computational cost.

In addition, this study highlights the potential of transfer learning as a powerful strategy for improving model adaptability across diverse molecular tasks. The ability of LLMs to generalize well across different datasets underscores the potential of transfer learning to enhance the versatility of machine learning models, regardless of the molecular representation's dimensionality. Future research should focus on extending this work by exploring tasks where 3D information might be more important e.g. protein docking. Investigating how BO performance scales with model size and complexity will also be crucial for optimizing these approaches. Additionally, the exploration of graph foundation models could open new avenues for representing and processing molecular data, combining the strengths of graph-based and large-scale foundation models.

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

# A  APPENDIX

## A.1  PSEUDOCODES

We present the pseudocode of the Bayesian Optimization (BO) loop and Section 5 in Algorithm 1.

---

**Algorithm 1** Using an NN as a feature extractor in BO.

---

**Require:** Pre-trained feature extractor $\phi_{W^*}$, mapping a molecular representation $c(x)$ to its embedding vector $h \in \mathbb{R}^H$; surrogate model $g_\theta : \mathbb{R}^H \to \mathbb{R}$; candidate molecules $D_{\text{cand}} = \{x_i\}_{i=1}^n$; initial dataset $D_1 = \{(x_i, f(x_i))\}_{i=1}^m$; time budget $T$.

1: **for** $t = 1, \ldots, T$ **do**
2:     $\Phi_t = \{(\phi_{W^*}(c(x)), f(x)) : (x, f(x)) \in D_t\}$
3:     $p(g_t|D_t) = \text{infer}(g_\theta, \Phi_t)$
4:     $x_t = \arg\max_{x \in D_{\text{cand}}} \alpha(p(g_t(c(x))|D_t))$
5:     $D_{t+1} = D_t \cup \{(x_t, f(x_t))\}$
6:     $D_{\text{cand}} = D_{\text{cand}} \setminus \{x_t\}$
7: **end for**
8: **return** $\arg\max_{(x,f(x)) \in D_{T+1}} f(x)$

---

## A.2  TRAINING

### A.2.1  FIXED-FEATURE SURROGATES

The following are the training details of the surrogates we used in Section 5. We used HuggingFace's transformers library (Wolf et al., 2020) for MolFormer. For GPs, we use BoTorch (Balandat et al., 2020) to construct the surrogate function. The Tanimoto kernel is taken from Gauche (Griffiths et al., 2024). To optimize the marginal likelihood, we use Adam (Kingma & Ba, 2014) with a learning rate of 0.01 for 500 epochs. We constrain the GNNs to $\tilde{1}.5$ million parameters, and further train Equifrormer v2 on noisy nodes. We optimize the GNNs with Adam with a learning rate of $1 \times 10^{-4}$ and weight decay of $5 \times 10^{-4}$ until convergence with early stopping at 20 epochs without improvement with a batch size of 64. We anneal the learning rate with the cosine annealing scheme (Loshchilov & Hutter, 2016). On the other hand for LLA, our implementation is based on the laplace-bayesopt package. The neural net used is a 2-hidden-layer multilayer perceptron with 50 hidden units on each layer along with the ReLU activation function. The Laplace approximation is done post-hoc, and we tune the prior precision with the marginal likelihood for 100 iterations. The Hessian is approximated with a Kronecker structure (Ritter et al., 2018).

## A.3  PROMPTING

Following the framework in Kristiadi et al. (2024), we used the prompt "The estimated {objective str} of the molecule {smiles str} is:" in our experiments. The variable `smiles_str` equals the SMILES representation of the molecule at hand, e.g., "OS(=O)(=O)O" for sulfuric acid. The variable `obj_str` has the value of the textual description of the problem at hand: "HOMO-LUMO gap in eV" for QM9, " atomization energy in kcal/mol " for QM7, "total energy in Hartees" for GEOM's Molecule Net and DRUGS.

