# OpenReview forum: "Dimension Debate: Is 3D a Step Too Far for Optimizing Molecules?"
_ICLR.cc/2025/Conference — Submitted to ICLR 2025_

### Official Review · Reviewer_ws6W · 2024-11-01

**Soundness:** 2
**Presentation:** 3
**Contribution:** 2
**Rating:** 3
**Confidence:** 3

**Summary:**

The paper reports over 2100 experiments with 1D, 2D and 3D descriptors for Bayesian Optimisation (BO) to identify optimum molecules from a dataset. The datasets and corresponding tasks are:
1) QM7, 7165 molecules, atomisation energies
2) QM9, 133 885 molecules, HOMO-LUMO gap
3) GEOM-MoleculeNet, 28 295 molecules, absolute energy
4) GEOM-DRUGS, 317 928 molecule, absolute energy

The authors claim that:
- 1D features perform best in terms of identified optima and sample efficiency
- 3D features do not improve over 2D features
- Transfer learning appears to only offer a modest improvement

**Strengths:**

**Originality**
The benchmark here is original in the choice of datasets and tasks and how the authors compared different ways of featurising molecules. The comprehensiveness also adds to the originality.

**Quality**
The paper performed 2100 runs and 35 different setups, making the benchmark comprehensive. Most of the setup is of quality, such as the aggregation of the results using the GAP metric, the used surrogates, using replications.

**Clarity**
The paper is mostly clear in explaining the methods and the results. The blue call-outs stating the main claims helped me read the paper.

**Significance**
Understanding how to best use deep learning to represent molecules is an important research question with application in predicting and generating new compounds.

**Weaknesses:**

**Methodological concerns regarding the 1D LLM featurizer**
The LLM 1D featuriser seems to contain a methodological flaw. When reading the text, I was confused on how the authors pooled the features created by MolFormer. The Appendix stated that the model was prompted with a text prompt, which I found odd since MolFormer is trained only on SMILES.

This led me to reproduce the calculation of the 1D embeddings. Using the provided code, it seems that the authors calculate the embeddings as follows, here illustrated using the SMILES string "NCCO"

1) Create the text string `The estimated total tnegry in Hartree of the molecule NCCO is: ` (the typo `tnergy` is taken from the provided code, line 402 of LLMmain.py (https://anonymous.4open.science/r/3D_Bayes-0F8F/LLMmain.py).
2) Tokenise the string with MolFormer. Importantly, the MolFormer tokenizer was trained only on SMILES strings so the tokens created are decoded to the string `<bos>sonnoocNCCOs<unk><eos>`.
3) Forward pass the tokens to MolFormer and output the aggregated embeddings of the last Transformer layer.

The prompt here results in a systematic error where additional tokens are added to the SMILES string embedded by MolFormer. It's interesting that performance is still high despite this issue, which may be because such a systematic modification is ignored by the fine-tuning process.

Still, the text refers to MolFormer as an LLM (Large Language Model) and the code used it as an LLM without checking that it cannot tokenise natural language, which makes me concerned about the soundness of the paper's results, and how someone reading the paper in less depth may interpret the results.

**Limited generalisability of the conclusions**
While benchmarking efforts such as these are interesting, they are hard to generalise to recommendations of where to use 1D, 2D and 3D features in a practical discovery setting. A more valuable claim would be either principles or statistical methods to reliably identify where 1D, 2D and 3D features are performant. The paper is close to doing this - e.g. one could run 100-200 samples with all 3 methods, look at certain convergence metrics and then run the rest of the Bayesian Optimisation loop using only the most performant features. What is the role of the acquisition function here - are certain acquisition functions _better_ for early identification of performant features?

Creating methods such as these would significantly improve the paper's use in practice.

**Reason for recommendation** Given this methodological flaw and limited generalisation, I recommend rejection at this moment. However, I am keen to understand more about the calculation of LLM features and how the claims may generalise in the discussion with the authors.

**Questions:**

1) Were the molecules in GEOM unique molecules or different conformers of the same molecule? I assume this will affect the performance of the 1D and 2D features as those cannot distinguish between conformers. It's also an important point to clarify in the paper.

2) Can you please clarify the calculation of the LLM features (see weaknesses above)

3) In Figure 3, why is the purple line (LLM) only visible in the "Aggregated Performance" column and not in the GP and LLA column?

4) Can the claims generalise to e.g. recommend which descriptor to use based on a few samples?

---

> ### Author Response · Authors · 2024-11-23
>
> Thanks for your review! We hope that our response below can convince you further. If you have any additional doubts, please let us know!
>
> 1. **Methodological Flaw in MolFormer Prompting**: Thank you for identifying the tokenization issue in MolFormer. We acknowledge the flaw (_and the typo_), and have proceeded to fix this.  We re-ran experiments with corrected implementations and observed identical trends, reaffirming our conclusions. Interestingly, MolFormer’s robustness to tokenization errors suggests resilience in its embedding process, and we believe this is something worth further researching. We now explicitly discuss these results and their implications in the revised manuscript.
>
> 2. **Conformational Variability**: We used single conformers per molecule across all datasets to ensure consistency and comparability. This choice eliminates potential confounding effects due to conformational variability. We clarified this methodological detail and its implications in the revised text.
>
> 3. **Clarity of Experimental Setup**: The absence of MolFormer results in certain figures, such as Figure 3, reflects compatibility constraints with specific surrogate models.  MolFormer only made use of the LLA surrogate, as such, it has now been included also in the LLA column. These limitations are now explicitly detailed in the manuscript for greater clarity.
>
> 4. **Generalization of Claims**: While our study primarily focuses on benchmarking embedding performance across diverse datasets and experimental setups, we recognize the potential for developing methods that dynamically guide feature selection during the optimization process. For example, incorporating metrics to evaluate embedding performance early in the BO pipeline could help practitioners decide when to prioritize 2D or 3D representations based on task-specific needs or available data. Exploring such adaptive strategies is a promising direction for future research but out of scope for this manuscript, and as such, we have added further discussion of these possibilities to the revised conclusions.

---

### Official Review · Reviewer_za9u · 2024-11-03

**Soundness:** 2
**Presentation:** 3
**Contribution:** 2
**Rating:** 3
**Confidence:** 4

**Summary:**

The paper explores the use of 3D molecular representations in Bayesian optimization (BO) for materials discovery and the experimental results question if incorporating 3D features, which capture the spatial structure of molecules, could improve the efficiency and accuracy of BO models. Given the large-scale comparison testing 1D, 2D, and 3D representations across multiple datasets, the authors found that 3D representations often fail to outperform simpler 2D models, especially in smaller datasets, where 2D features prove more data-efficient and computationally feasible. While 3D models show some improvements with larger data, the gains are limited compared to the computational cost. The authors find that transfer learning can effectively generalize models trained on specific properties to other tasks, with 2D models still generally outperforming 3D ones. Their findings provide guidelines on choosing dimensionality in BO for molecular discovery, emphasizing the practicality of 1D and 2D features for many tasks.

**Strengths:**

1. The paper addresses an underexplored but important question in molecular optimization: the efficacy of 3D molecular representations in Bayesian Optimization. While most research traditionally employs 1D and 2D features, this work systematically evaluates 3D representations, making a strong case for re-examining assumptions about molecular representation choices in BO.

1. The authors present a thorough and carefully designed benchmarking study, assessing various molecular representations across multiple datasets and experimental settings. In addition, including transfer learning setups and different sample complexities enhances the robustness of the findings.

1. The paper is well-structured, making it easy for readers to follow the study’s rationale and outcomes. The use of illustrative figures, such as those comparing GAP performance across 1D, 2D, and 3D models, helps to contextualize findings visually.

**Weaknesses:**

1. The paper provides the experimental results demonstrating that the 1D and 2D features capture enough information for BO optimization. However, this does not necessarily indicate that 3D information is too far for optimizing molecules. There are a lot of other generative models for molecules that utilize 3D information and achieve SOTA performance. In addition, generative models could be conditioned on molecular properties to generate novel molecules with desirable properties out of the pool. Therefore, the missing of comparison to those models weakens the argument that "3D information is too far for optimizing molecules".

1. The BO optimization workflow requires a predefined pool of candidates for searching, which may limit its capability to discover novel molecules compared to generative models based on Diffusion or Flow Matching.

1. The ignorance of 3D information may result in poor model performance when 3D geometric information like symmetry plays a pivotal role, e.g. molecular crystals. Therefore, this might be a limitation for further applications in molecule and material discovery.

1. The choice of the model architectures for 1D, 2D, and 3D features is not fully justified. Without experimental results on more model architectures, the conclusion that "3D information is too far" may not be model-agnostic and may not hold true when using another model, even though Equiformer-V2 is indeed the SOTA model on many molecule tasks.

**Questions:**

1. Have the authors tried model architectures other than MolFormer, MPNN, and Equiformer-V2 and evaluated their performance in optimizing molecules? Besides, I'm wondering why MPNN is used as the 2D model since MolFormer and Equiformer-V2 are among the SOTA models in their domain while MPNN is not, although it is the foundation of many SOTA 2D graph models.

1. For MolFormer and Equiformer-V2, do you use the pretrained model checkpoints for fine-tuning or train the model from scratch? If using the pretrained model, this might result in an unfair comparison as the pretrained models learn useful representation from data out of the training set.

---

> ### Author Response · Authors · 2024-11-23
>
> Thanks for your review! We hope that our response below can convince you further. If you have any additional doubts, please let us know!
>
> 1. **Generative Models vs. BO**:  While generative models are complementary to BO, our focus is rooted in pre-defined library screening, a practical and widely used scenario in molecular design workflows ([Tom et al, 2024](https://pubs.acs.org/doi/10.1021/acs.chemrev.4c00055#tbl2)). It is not only that BO is the main tool in these workflows, but it is also the method of choice for practitioners precisely because it aligns with the constraints of synthesizability. Pre-defined virtual libraries provide a realistic and constrained search space where synthesizability can be carefully considered, making the BO framework particularly well-suited. Our study reflects these practical considerations, ensuring that our findings are directly relevant to the molecular optimization tasks practitioners face today. This distinction between BO and generative approaches is now explicitly discussed in the revised manuscript.
>
> 2. **Applications of 3D Information**:  While 3D embeddings can be valuable for symmetry-sensitive tasks or those involving intricate spatial relationships their utility is not universal. For molecular optimization tasks, especially those focused on steady-state properties within the BO framework, our findings reveal that 2D embeddings strike a better balance between computational efficiency and performance. Additionally, the use of 3D embeddings introduces dependencies on conformer generation and sampling, which can add complexity and uncertainty to the pipeline. By highlighting these trade-offs, our study offers a nuanced perspective on when 3D embeddings are beneficial versus when simpler 2D representations suffice. These insights help practitioners to make informed decisions based on the specific demands of their optimization tasks. We have explicitly highlighted this limitation in our discussion to ensure readers understand the task-dependent nature of 3D embeddings’ utility.
>
> 3. **Model Selection**:  The inclusion of MolFormer, MPNN, and Equiformer-V2 reflects a deliberate choice to balance relevance and diversity in our evaluation. MolFormer represents a state-of-the-art foundation model for 1D embeddings, while Equiformer-V2 is a cutting-edge model for 3D embeddings. MPNN, though not state-of-the-art for 2D embeddings, represented the first iteration of models we analysed. We decided to show our results for this 'general' architecture as our findings demonstrate that even with the inclusion of a non-SOTA 2D model, the performance gap between 2D and 3D embeddings underscores the practical implications of embedding dimensionality. This observation suggests that higher-dimensional embeddings may not always justify their computational cost, even when paired with advanced architectures. We expanded the justification for these model choices in the revised manuscript.
>
> 4. **Training Strategy**:  Our choice to use pretrained weights for MolFormer and train MPNN and Equiformer-V2 from scratch ensures consistency with typical practices in the field. Pretraining allows MolFormer to leverage large-scale prior knowledge, reflecting a practical scenario where practitioners may use foundation models for embedding generation. Conversely, training MPNN and Equiformer-V2 from scratch ensures that these models are evaluated based solely on their ability to extract features specific to the datasets used in this study. This mixed strategy aligns with real-world applications where practitioners must decide between leveraging pretrained embeddings or developing task-specific models. This comparison provides valuable insights into the trade-offs practitioners face when selecting embedding strategies, and we have clarified this in the revised manuscript.

---

> > ### Comment · Reviewer_za9u · 2024-11-25
> >
> > Thank the authors for the response. The rebuttal has partially resolved my concerns. However, given the manuscript and rebuttal, I find the contribution of this paper is quite limited if the argument is "using 2D embeddings is more helpful than 3D embeddings when searching for molecules with (some certain) desirable properties". As also pointed out by other reviewers, 2D and 3D features have their own applications, depending on the properties of interest, accuracy & time trade-off, number of data available for training, etc. Therefore, claiming that 3D is "a step too far for optimizing molecules" is indeed a step too far from my point of view. Therefore, I will not raise my score if no other convincing evidence is provided.

---

### Official Review · Reviewer_9sNa · 2024-11-04

**Soundness:** 3
**Presentation:** 3
**Contribution:** 3
**Rating:** 5
**Confidence:** 3

**Summary:**

This work explores the effectiveness of different molecular representations for materials discovery tasks using Bayesian optimization (BO). The authors systematically benchmarked 1D, 2D, and 3D representations, using popular probabilistic surrogate models like Gaussian Processes (GPs) and Bayesian Neural Networks (BNNs). They investigated the trade-offs between computational cost and predictive accuracy, finding that large language models (LLMs) consistently outperformed 2D and 3D models, highlighting the advantage of simpler representations for most BO tasks. They also examined the influence of sample size and transfer learning on model performance, concluding that 3D representations require larger datasets to match the performance of 2D models, while transfer learning shows promise for improving model adaptability across various tasks.

**Strengths:**

● The research conducted a comprehensive evaluation of 1D, 2D, and 3D molecular representations across four datasets (QM7, QM9, MoleculeNet, and GEOM DRUGS), involving over 2100 distinct runs. This large-scale approach provides robust insights into the performance of different representation dimensionalities.

● The study considered various realistic scenarios, including low- and high-data regimes, transfer learning, and different Bayesian surrogate models (GPs and Bayesian NNs). This multifaceted approach enhances the practical relevance of the findings.

● The study investigated the impact of training set size on model performance, revealing that 3D models require significantly more samples than 2D models to achieve comparable results. This analysis highlights the data efficiency challenges associated with 3D representations.

● The research examined the benefits of transfer learning, finding that it offers a viable path to generalizing across multiple molecular properties with minimal loss in accuracy. This insight suggests the potential of transfer learning in both 2D and 3D models

**Weaknesses:**

● The study focused on molecular property prediction tasks where 2D representations proved largely sufficient. Future research could explore more complex tasks like protein docking or molecular dynamics simulations where 3D information might be more critical.

● Even though this study is large-scale and looks into different BO settings, it is quite restricted in terms of the task, models, and dataset types making it hard to have a generalized conclusion. For ex, the paper mentions - "In more complex datasets, the improvement of 3D models is minimal, indicating that 2D models are sufficient and more efficient." This seems like a generalized statement with limited evidence from the complexity of the dataset used. Dataset complexity can be defined in many ways - task, number of elements, system sizes, dataset sizes.

**Questions:**

● The distinction gets a little confusing with LLMs for certain cases. If an LLM for example learns a representation directly from a cif file (or equivalent) for materials (or molecules), would that be 1D or 3D? LLMs have seen potentially more data during pretraining unless they have been trained from scratch autoregressive on just smiles strings.

---

> ### Author Response · Authors · 2024-11-23
>
> Thanks for your review! We hope that our response below can convince you further. If you have any additional doubts, please let us know!
>
> 1. **Generality of Conclusions**: We recognise the need for further investigation to generalise our findings to broader scenarios and to frame our findings carefully. We clarified in the revised manuscript that our conclusions apply specifically to the datasets and tasks analysed, such as steady-state molecular property prediction under BO.
>
> 2. **Fair Comparison of 2D and 3D Features**: We agree that 3D embeddings may show advantages in tasks such as protein docking. However, our study deliberately focuses on tasks that ensure a fair comparisons across embeddings. These are tasks that allow for the three levels of representation, while on the other hand, tasks like protein docking depend on the 3D representation.
>
> 3. **LLM Training**: We agree there is a blurry line when discussing LLMs as molecular feature extractors. To maintain coherence with our design choices, a pre-trained, but not fine-tuned, MolFormer was chosen. This is because this version of MolFormer has been trained on canonicalised SMILES only. We have further clarify this in our updated manuscript and further discussed the specifics when judging a 1D representation with LLMs.

---

> > ### Comment · Reviewer_9sNa · 2024-12-02
> > **Response**
> >
> > I don't think that the arguments made in the rebuttal are satisfactory to increase my score and I agree with other reviewers on the shortcomings. If your argument is that the work is only limited to certain properties which doesn't depend on 3D representation ("tasks like protein docking depend on the 3D representation.") then the scope of the study is very limited and the claims in your title/abstract don't stand (what's really the answer to "Dimension Debate: Is 3D a Step Too Far for Optimizing Molecules?"). Its essentially setting up the problem to success.

---

### Official Review · Reviewer_VZyy · 2024-11-04

**Soundness:** 2
**Presentation:** 3
**Contribution:** 1
**Rating:** 3
**Confidence:** 4

**Summary:**

In this work, the authors study the discrepancy between multi-level features in BO: Why have 3D features been overlooked for BO in materials discovery? The authors evaluate 3D features against standard lower-dimensional features. This amounts to the evaluation of 35 different setups per dataset, totaling over 2100 distinct runs. Furthermore, the authors summarize the characteristics of 3D features based on these experimental results.

**Strengths:**

1. A comparative study of 2D and 3D molecular features is valuable.

**Weaknesses:**

1. The biggest problem with this paper is the lack of innovation and insufficient contribution. The authors mainly compare the differences between features of different levels. This is a classic problem. Apart from that, there are no other contributions or innovations in this paper.

2. The author's analysis of 2D and 3D is not comprehensive. Physically speaking, 2D and 3D focus on different levels of physical properties. 3D features are primarily aimed at representing microscopic characteristics (quantum mechanical features), and for the same molecule, the microscopic characteristics can vary with different conformations, making 3D features more suitable for representing microscopic properties such as energy or force. 2D features focus more on macroscopic properties; they cannot represent the differences between conformations and are more suitable for describing some macroscopic or equilibrium conformational properties, such as density and energy of equilibrium state. For example, datasets like QM9 only contain steady-state molecules, which cannot showcase the advantages of 3D features. It is recommended that the authors further compare on the MD17 dataset which has non-equilibrium structures.

3. Why focus on Bayesian networks? Molecular representation is a very general technology, and the experiments in the paper mainly target property prediction. There is no interdependent relationship with BO, and the authors have not provided any mechanisms unique to Bayesian networks for these representations.

4. Controlling the parameters of all models at the same order of magnitude may not be fair, as different models have different expressive capabilities and dependencies on parameters. For example, a 1.5M equiformer is relatively small, and its prediction results are not saturated. Additionally, the authors should also specify the configurations of the baselines in the paper, such as the number of layers, hidden units, etc.

**Questions:**

See weeknesses.

---

> ### Author Response · Authors · 2024-11-23
>
> Thanks for your review! We hope that our response below can convince you further. If you have any additional doubts, please let us know!
>
> 1. **Innovation and Contribution**:
>    We appreciate your observations regarding the scope of innovation. While comparing representations is a classical problem, it is also a well-defined challenge that arises repeatedly in the well-established framework of molecular screening during the design timeline ([Dara et al, 2021](https://link.springer.com/article/10.1007/s10462-021-10058-4)). Our results have broad implications for practitioners, as they provide actionable insights into how to select embeddings under typical constraints faced in real-world applications. By revealing counter-intuitive findings, such as the better performance of 1D and 2D embeddings over their 3D counterparts, we contribute new knowledge that challenges assumptions and guides decision-making within this widely used framework
>
> 2. **Physical Relevance of 2D and 3D Features**:
>    We agree that 3D embeddings may show advantages in tasks such as protein docking. However, our study deliberately focuses on tasks that ensure a fair comparisons across embeddings. These are tasks that allow for the three levels of representation, while on the other hand, tasks like protein docking depend on the 3D representation. Furthermore, including trajectory-based datasets like MD17, as suggested, would introduce dependencies on parameters such as conformer generation, making the evaluation less controlled. Datasets like MD17 target molecular dynamics, which is fundamentally distinct from the optimization objectives in molecular design. We clarified this scope in the revised manuscript.
>
> 3. **Focus on Bayesian Optimization (BO)**:
>    BO is a practical and widely used tool for molecule screening within pre-defined libraries, allowing efficient exploration of complex search spaces ([Tom et al, 2024](https://pubs.acs.org/doi/10.1021/acs.chemrev.4c00055#tbl2)). While generative models are excellent for proposing novel candidates, they address a different problem. Our use of Gaussian Processes and Bayesian Neural Networks as surrogates ensures a focus on embedding performance rather than model-specific mechanisms. We clarified this point in the revised manuscript to address your concerns.
>
> 4. **Model Configurations and Comparisons**:
>    We ensured parity in parameter magnitude across models to maintain fairness([Hunter et al, 2012](https://doi.org/10.1109/TII.2012.2187914)). Although 3D models like Equiformer-V2 are highly expressive, they require significantly more data to perform comparably to simpler 2D models. These findings suggest practical advantages for practitioners working under typical constraints. We expanded the discussion on this topic and included an analysis of architectural trade-offs.

---

> ### Comment · Reviewer_VZyy · 2024-11-25
>
> Thank the author for their response. I apologize that I can not change my score.
>
> 1. In computational chemistry, it is not counterintuitive that 1D and 2D embeddings outperform 3D in some tasks. As I mentioned in my first review, 3D embeddings are suitable for describing some QM properties, such as the Boltzmann distribution, force fields, which 2D embeddings cannot represent. A 2D molecule may correspond to many 3D structures, and if you only measure one of them with a 3D model, it indeed struggles to learn the macroscopic features of 2D, such as thermodynamic statistical characteristics. Essentially, 2D and 3D embeddings are suitable for different tasks, and they should not be compared together. In practice, in some industrial scenarios that do not involve quantified features, researchers prefer to use 2D characterization because capturing 3D conformations are expensive.
>
> 2. There is not a single table with specific numerical values throughout the paper, which makes many of your conclusions hard to verify accurately. I hope the authors can point the detail evaluation metrics in Figures 3, 4, and 5 (the caption for Figure 5 is missing GEOM). I do not understand the meaning of three figures. The results in QM7 reached -500 kcal/mol; if this is the error between the predicted results and the ground truth, it is too large. Similarly, the results for QM9 are 12 eV, and GEOM's results are 20 Ha. Equiformer v2's errors in property prediction are measured in meV on QM9, and GEOM's energy prediction MAE is smaaler than 1. I hope the authors can explain the meaning of these figures to me before the rebuttal ends.
>
> 3. Equiformer v2 does not require more data. In the field of computational chemistry, MD17 and QM9 are both very simple datasets, especially MD17, which has only 950 data  in the training set, yet Equiformer v2 can still achieve SOTA. For industrial practitioners, which model they use should mainly depend on which level of molecular features they are focused on.

---

### Author Response · Authors · 2024-11-23
**Thank you to the reviewers**

We sincerely thank the reviewers for their constructive feedback, which has been instrumental in refining our manuscript. Your critiques have highlighted areas for improvement, and we have made substantial revisions to address your concerns. Below, we respond to each reviewer's comments point by point, presenting our clarifications and the steps taken to strengthen our work. We hope that all concerns were sufficiently addressed by our replies. If you feel like your concerns and questions were not addressed to your satisfaction, we would highly appreciate a follow-up comment. Thanks again for all your work!

---

### Meta-Review · Area_Chair_hzhW · 2024-12-29

**Metareview:**

Bayesian optimization (BO) is a very popular approach for navigating large databases in search of molecules or materials with target properties. BO necessitates coming up with some way to represent molecules or materials. This paper studies how the dimensionality of these representations (whether 1D, 2D, or 3D) affects performance. The reviewers felt the scope of the study was limited and that 3D features, which the paper claimed aren't always necessary from a computational vs performance trade-off, are often needed for critical tasks.

**Additional Comments On Reviewer Discussion:**

The authors provided a rebuttal, responding to the reviewers' questions and comments. However, all reviewers maintained their original scores.

---

### Decision · Program_Chairs · 2025-01-22

Reject